# Essential newborn care practices and associated factors among home delivered mothers in Guto Gida District, East Wollega zone

**Mulugeta Abebe[1], Gemechu Kejela[2]\*, Melese Chego[2], Markos Desalegn[2]**

**1** Department of Nursing, Wollega University Referral Hospital, Institute of Health Sciences, Wollega University, Nekemte, Ethiopia, **2** Department of Public Health, Institute of Health Sciences, Wollega University, Nekemte, Ethiopia

\* gemechukejela86@gmail.com

## Abstract

### Background

Essential Newborn Care is a set of strategic and cost-effective interventions planned to improve the health of newborns through the care they receive from birth up to 28 days. In the current study area, little is known about the prevalence of essential newborn care practices and its associated factors. This study was aimed to assess the prevalence of essential newborn care practice and its associated factors among home-delivered mothers in Guto Gida district, western Ethiopia.

### Methods

A community-based cross-sectional study was conducted in Guto Gida district from September 5 to 15, 2020. Data were collected by interviewing 601 systematically selected home-delivered women. Descriptive statistics were employed to describe frequency and percent. Binary logistic regression analysis was employed to identify candidate variables for the final model. Variables with p-value less than 0.25 at bivariate logistic regression were considered as the candidate variable and entered into multivariable logistic regression model. Finally, multivariable logistic regression was employed to identify associated factors at p-value less than 0.05, and the strength of association was described by adjusted odds ratios with 95% CI.

### Results

The study shows that the level of essential newborn care practices was 168 (28%) (23.9–31.4). In this study, women in the first wealth quantile (AOR [95% CI] = 0.64 [0.34–0.97]), women who had one live birth (AOR [95% CI] = 0.51 [0.22–0.87]), women who lost their neonate before the study period (AOR [95% CI] = 0.11 [0.05–0.22]) were less likely to practice essential newborn care. Women who were advised on essential newborn care practice during a home visit by health extension workers (AOR [95% CI] = 3.45[1.56–7.26]), women

**Data Availability Statement:** All relevant data are within the paper and its Supporting Information files.

**Funding:** The authors received no specific funding for this work.

**Competing interests:** The authors have declared that no competing interests exist.

who attended antenatal care during their current pregnancy (AOR [95% CI] = 1.79 (1.21–3.36]), and women who were attended at their birth by health extension workers (AOR [95% CI] = 3.29 [2.13–5.94]) were more likely to practice essential newborn care.

## Conclusions

In this study, the prevalence of essential newborn care practice was low (28%), as compared with the World Health Organization recommendation that it should be 100%. The wealth quantile, number of live births, home visits by health extension workers, antenatal care, birth attendant, and neonatal death were independent predictors of essential newborn care practices.

## Introduction

The first 28 days of life have been the most vulnerable time for newborn survival, particularly the first day, the first week, and the first month of life have been the most critical times for the survival of children [1–3]. Essential Newborn Care, which is a set of strategic and cost-effective interventions planned to improve the health of newborns through the care they receive from delivery till 28 days of life is crucial for their survival. These interventions are: cutting the cord by using a new or boiled blade; tying the cord by using new or washed thread; and preventing the application of any substance to the cord stump [4]. Wrapping and drying the newborn immediately after birth, delaying the first bath for 24 hours or more, early initiation of breast-feeding within one hour of birth, providing first milk (colostrum) for the newborn, and preventing pre-lacteal feeding [4–6].

Every year, over 130 million babies are delivered worldwide; 40 million of them are delivered at home without the help of a skilled birth attendant, which has a great risk for mothers and newborns. In Ethiopia, approximately 2.6 million births occur annually. Of these, 73%, or 1.83 million, occur at home without the help of a skilled birth attendant, which contributes to the low utilization of Essential Newborn Care Practice (ENCP) [5–9].

Despite proven cost-effective solutions to reduce neonatal mortality, there was a low utilization of ENCP and relatively little improvement in neonatal mortality compared to under-five mortalities. Low and middle income countries account for nearly all neonatal deaths. Ethiopia is one of the countries with high neonatal mortality, along with India, Nigeria, Pakistan, Somalia, and the Democratic Republic of the Congo. This mortality occurred as a result of complications related to poor care during postnatal and delivery [10–12].

To increase this low utilization level of ENCP to 100% as recommended by the World Health Organization (WHO) and to prevent causes of neonatal mortality, WHO, Ethiopian Federal Ministry of Health (EFMOH), different scholars and many other stakeholders have provided guidelines, allocated budgets and conducted multiple activities that encompass cleanliness of institutional delivery, thermal protection, breast-feeding and cord care [11, 13–15].

Despite all the above efforts, neonatal mortality is still very high in the country. To the best of the researcher's knowledge, little is known about the level of ENCP and its associated factors in the study area. In addition to this, 2020 Health Management Information System (HMIS) report for Guto Gida district showed that the majority of births (52% of births in the district) occurred at home and utilization of essential newborn care practice was low in the district [16]. Therefore, this study aimed to assess the prevalence of essential newborn care practice and identify its associated factors among home-delivered mothers in Guto Gida district.

## Methods and materials

### Study design and setting

A community-based cross-sectional study was conducted from September 5 to 15, 2020, among home-delivery mothers in Guto Gida district, East Wollega zone of Oromia region. According to Guto Gida district map that was obtained from the district administrative office, the district is at an altitude of 905'N 36,033'E, and longitude 9,083,036'E with an elevation of 2,088m and is divided into twenty-one rural kebeles (the lowest administrative unit in Ethiopia). According to information obtained from the Guto Gida district administrative office, there are 21 rural health posts (two health extension workers were assigned to each health post), 3 health centers (lead by BSc Nurse or public health professional), and no hospital in the district. According to the 2007 Population and Housing Census report of Ethiopia, the district had a total population of 1,19101, of which 57,168 were men and 61,933 were women, and there were a total of 27,068 households and 26,927 women in the reproductive age group (15–49 years). The average number of households in one kebele was 1,289, with a total population of 5,671 [16].

### Population

All women who delivered at home and permanent residents in Guto Gida district were the source population, and all home-delivered women in five randomly selected kebeles of Guto Gida district were the study population. Women in the reproductive age group (15–49), who resided in the study area for at least six months, who were reported to have delivered an alive baby within six months of postpartum and who might have had an infant loss prior to the study period were included into the study. Mothers who were seriously ill, had a history of mental illness, and were unable to care for their infants, or care givers/guardians were excluded from the study.

### Sample size determination and sampling procedure

The required sample size of the study was determined by using the single population proportion formula based on the following assumptions: A prevalence of essential newborn care practice of 24% from a study conducted in Damot Pulasa district, south Ethiopia [9], 5% margin of error, a 95% confidence level, design effect of 2 (since the study employed multistage cluster sampling), and 10% for non-response. Depending on all these assumptions, the final sample size becomes **617.**

A multi-stage sampling was employed to perform the procedure. First, five kebeles were selected from 21 kebeles in Guto Gida district, by lottery method. The number of all home-delivered women within the six months in those five selected kebeles were identified by consulting health extension workers of the respective kebeles. Then, the determined sample size was proportionally allocated to each selected kebele. The study participants were then selected from all selected kebeles by systematic random sampling, after calculating sampling interval as follows: K = (N/n) = [2495/617] = 4.

### Measurements

Essential newborn care practice was good and coded as 'Yes', for mothers who were practiced three domains (safe cord care, optimal thermal care, and good neonatal feeding) appropriately whereas the practice was poor and coded as 'No', if one component was missed from the three domains [4, 10, 17]. Safe cord care was defined as the use of a clean cutting instrument to cut the umbilical cord (boiled new, used blade or scissor) plus clean thread, cord tie or cord clamp

and no any substance applied on the cord stump [4, 10, 17]. Thermal care was defined as wiping off/drying the baby within ten minute, wrapping in new or clean and dry old cloth and washing the body of the newborn by warm water after 24 hour of delivery to prevent hypothermia [4, 10, 17]. Good neonatal feeding was defined as initiating breastfeeding within the first one hour after birth, giving no prelacteal feeding, and feeding the child with colostrum [4, 10, 17].

Wealth index was assessed by using Ethiopian Demographic and Health Survey household assets questions and analyzed by using principal component analysis. The wealth index in this study was ranked into three categories as: 1st quintile, 2nd quintile and 3rd quintile [4, 10]. Knowledge about ENCP was good for mothers who responded correctly to greater than 50% of knowledge related questions. Whereas, knowledge was poor for mothers who responded correctly to less than or equal to 50% of knowledge related questions [4, 10]. Knowledge of mothers on neonatal danger signs was good when mothers identified at least four neonatal danger signs among the six common neonatal danger signs [4, 10]. The six common neonatal danger signs in this study were; poor sucking or not able to feed breast, fast breathing, hypothermia, fever, drowsy or unconscious, cord bleeding and infection [11, 18].

## Data collection tools and procedures

A structured interviewer administered questionnaire developed from previous studies [5, 9–11] was used to collect the data. The questionnaire was developed in English and then translated into the regional language (Afan Oromo). The Afan Oromo questionnaire was retranslated to English by an individual who is fluent in both languages to check its consistency and better understanding by both data collectors and respondents. The questionnaire contains sociodemographic questions, service utilization related questions, and maternal knowledge related questions. Supervisors and data collectors modified the questionnaire based on pre-test findings, local traditions, and cultural sensitivity. Clarities were also made on vague questions before actual data collection. Finally, the Afan Oromo version of the questionnaire was utilized for actual data collection.

Four grade 12 completed data collectors collected the data and one BSc Nurse supervised data collection process. In addition, one health extension worker was recruited to show the geographical locations of home-delivered mothers. One-day, appropriate training was given for data collectors and supervisors before the data collection. Training was conducted on the purpose and objectives of the study, the role and responsibility of data collectors and supervisors, the content and meaning of each question, the way of respondent approach, and how to conduct the interview.

## Data quality assurance

In this study, data quality was assured by using different approaches. The English version of the questionnaire was translated into Afan Oromo for ease of fieldwork, and then it was translated back to English to check its consistency. Training and orientation were provided for all data collectors and supervisors. The questionnaire was pre-tested at Uke kebele (the kebele out of the study area) on 40 (5%) mothers who have similar characteristics with the study population, five days before actual data collection. During data collection, the supervisors conducted onsite supervision, and every evening, they checked the accuracy, consistency, and completeness of all filled questionnaires. Incomplete questionnaires were returned to data collectors to complete or to recollect again. The supervisors were re-interviewed 5% of the participants. In addition, double data entry was also made to check its consistency.

## Data processing and analysis

After checking its completeness, the collected data were coded, cleaned, and entered into Epi-Data version 3.1 and exported to SPSS for Windows version 24. Descriptive statistics were done to show frequency and percentage distribution of the variables. A binary logistic regression was conducted to check the significant relationship between dependent and independent variables. Variables that had a P value of less than 0.25 in binary logistic regression were considered as the candidate variables for the multivariable logistic regression. A step-wise backward logistic regression was used to identify the association between dependent and independent variables. The strength of association between multiple independent variables and a dependent variable was declared based on their adjusted odds ratio (AOR) with 95% CI. Finally, variables with p-value <0.05 at the final model were considered as statistically significant predictors of the outcome variable. Before running multivariable logistic regression analysis, the multi-co-linearity effect was cheeked by using variance inflation factor (VIF). As a result, all variables had a VIF of <10, indicating no collinearity problem. Model fitness was also checked by Hosmer-Leme show goodness of fit. The result of the Hosmer-Lemeshow showed insignificant result, indicating the model was fit for the analysis.

## Ethical considerations

Ethical clearance was obtained from Research Ethics and Review Committee (RERC) of Wollega University. Then, letter of cooperation was written to Guto Gida district administration. Finally, the aim of the study was explained and informed written consent was obtained from the study participants.

## Results

### Demographic and socio-economic characteristics

Out of the total 617 calculated sample size, 601 women participated, making the response rate of 97.6%. Their age ranged from 16 to 41 with a mean of 25.76 (+3.786 SD) years. The majority 281 (46.8%) of participants belonged to the age group of 25–34 years. The majority of the study participants, 542 (90.2%), were Oromo, and more than four-fifth of the respondents, 522 (86.9%) were Protestant.

Almost all, 586 (97.5%) of the study participants were married. About two thirds, 408 (67.9%) of the respondents did not attended formal education, while 162 (27.0%) attended primary school and only 31 (5.2%) attended secondary school. Furthermore, 589 (98.0%) of the respondents were housewives, and about 354 (58.9%) of the respondents belonged to the second wealth quintile (Table 1).

### Maternal factors and neonatal condition

From the total (601) study participants, majority of them 483 (80.3%) delivered their first baby at the age of > 18 years, and more than half 339 (56.4%) of the respondents had 2–4 children. Only 28 (4.7%) of participants reported that their neonates died before the study period, with half of these neonates dying due to a lack of ENCP and 11 (39.3%) dying on the first day of life.

### Reasons for home delivery

From the total 601 women interviewed, 574 mothers had reason for their home delivery as; 119 (20.7%) because of no nearby health facility, 181(31.5%) due to lack of transportation, 318 (55.4%) due to precipitated labor, and 185 (32.2%) were due to lack of money for service (Fig 1).

**Table 1. Socio-demographic and economic characteristics among home delivered women, in Guto Gida district, Oromia, September 2020 (n = 601).**

| Variables | Category | Frequency | Percentage |
|---|---|---|---|
| Mothers age group | 15–19 | 83 | 13.8 |
| | 20–24 | 94 | 15.6 |
| | 25–29 | 140 | 23.3 |
| | 30–34 | 141 | 23.5 |
| | 35–39 | 98 | 16.3 |
| | 40–44 | 45 | 7.5 |
| Mothers Religion | Protestant | 522 | 86.9 |
| | Orthodox | 48 | 8.0 |
| | Muslim | 31 | 5.2 |
| Mothers Ethnicity | Oromo | 542 | 90.2 |
| | Amhara | 52 | 8.7 |
| | Tigre | 7 | 1.2 |
| Mothers Marital status | Married | 586 | 97.5 |
| | Others (single, widowed and divorced) | 15 | 2.5 |
| Occupation | Housewife | 589 | 98.0 |
| | Others (merchant, labor work and student) | 12 | 2 |
| Wealth index in quintile | 1st quintile, | 187 | 31.1 |
| | 2nd quintile | 354 | 58.9 |
| | 3rd quintile | 60 | 10.0 |

## Maternal health service utilization

The majority, 453 (75.4%) of respondents were visited and advised at their home by HEWs during pregnancy and PNC. More than four-fifths 515 (85.7%) of the study participants received ANC services, and 108 (18%) of them received their first ANC visit at less than 4 months of gestation.

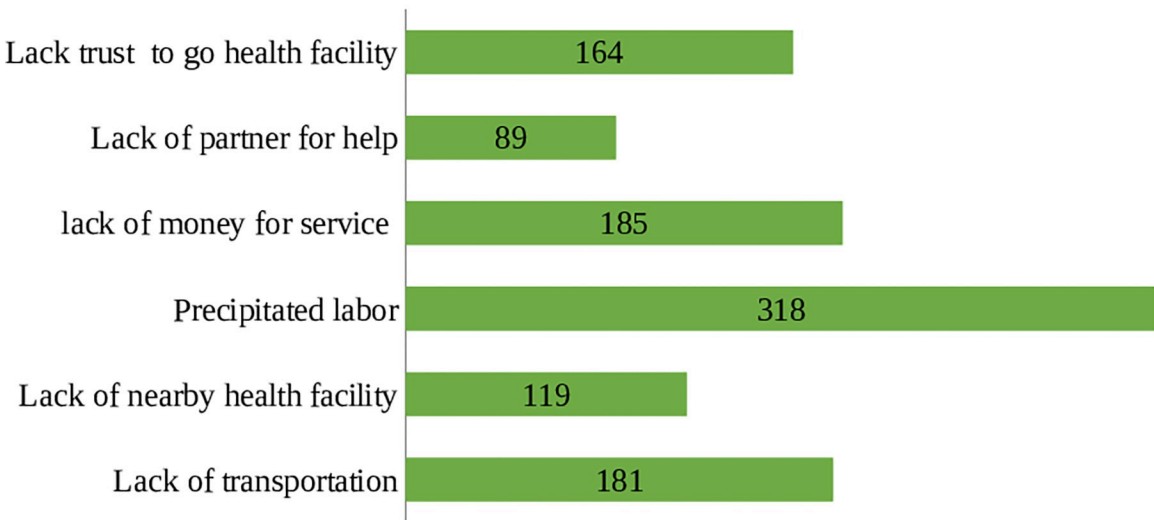

**Mothers Reasons for their home delivery(n= 574)**

- Lack trust to go health facility: 164
- Lack of partner for help: 89
- lack of money for service: 185
- Precipitated labor: 318
- Lack of nearby health facility: 119
- Lack of transportation: 181

**Fig 1. Reasons of postpartum women for their home delivery, in Guto Gida district, Oromia regional state, Ethiopia, 2020, (n = 574).**

From the total 515 participants who received ANC services, 194 (32.3%), 226 (37.6%), and 35 (5.8%) of them made one visit, two visits, and three visits, respectively. However, only 60 (10%) of participants received each of the four sets of focused ANC services.

About 499 (83.0%), 292 (48.6%), and 58 (9.7%) of all study participants received some ANC services from HEW, midwife/nurse/HO, and doctor, respectively. One hundred sixty-five (32.0%) of participants received their entire ANC visit only from HEWs, and 16 (3.1%) of participants received their entire ANC visit only from other health workers, other than HEWs.

With regard to the persons attended their birth, 330 (54.9%) of them were attended by traditional birth attendants (TBA), 139 (23.1%) of them were attended by HEWs, and 132 (22% of them were attended by others (mother/sisters/other-relatives/friends).

Only 160 (26.6%) of participants visited their PNC. Out of those who visited their postnatal care, 95 (59.3%), 34 (21.3%), and 31 (19.4%) of them visited their PNC once, twice, and three times respectively. Regarding the date of first PNC attendance, 15 (2.5%), 63 (10.5%), 30 (5.0%), 25 (4.2%), and 27 (4.5%) of women attended their first PNC in less than 4 hours, within 4–23 hours, within 1–2 days, within 3–6 days, and within 7–41 days respectively (Table 2).

## Knowledge of mother on ENCP

From the total study subjects (601), 560 (93.2%) of women had an awareness/information on ENCP. Four hundred ninety (87.3%) of women were responded the cord of the newborn were cut by using new razor blade, 514 (91.8%) of women were responded the cord were tied by new string or thread and 401 (71.6%) of them were responded nothing applied to the cord of newborn from birth up to 7 days. Three hundred fifty-two (62.9%) of women were responded as the newborn were washed/bathed after one hour. Four hundred eleven (73.4%) of women were responded as the newborn were breast fed within one hour after birth and 499 (89.1%) of them were responded as the newborn baby should feed the first breast milk/colostrum. Of those, who responded to the knowledge questions, 454 (81.1%) correctly answered more than 50% of the questions, and they were classified as having good knowledge of ENCP (Table 3).

## Knowledge of mothers on neonatal danger signs

Out of the total respondents (601), 477 (78.4%) had an information/awareness on neonatal danger signs. From those women, 247 (51.8%) correctly listed 4 or more than 4 neonatal

**Table 2. Health service utilization among home delivered women, in Guto Gida district, Oromia regional state, Ethiopia, September 2020 (n = 601).**

| Variables | Category | Frequency | Percentage |
|---|---|---|---|
| Home visit by HEW during current pregnancy and after delivery | Yes | 453 | 75.4 |
| | No | 148 | 24.6 |
| Advised by HEWs on hand washing with soap and water before care | Yes | 342 | 56.9 |
| | No | 111 | 18.5 |
| Advised by HEWs on drying and wrapping the neonate immediately after delivery | Yes | 124 | 20.6 |
| | No | 329 | 54.7 |
| Advised by HEWs on breastfeeding immediately after birth within 1 hour | Yes | 333 | 55.4 |
| | No | 120 | 20.0 |
| Advised by HEWs on Neonatal danger sign | Yes | 346 | 57.6 |
| | No | 107 | 17.8 |
| Attended ANC by any health professional during current pregnancy | Yes | 515 | 85.7 |
| | No | 86 | 14.3 |
| Travelled to health institution for PNC | Yes | 160 | 26.6 |
| | No | 441 | 73.2 |

**Table 3. Knowledge status of home delivered women toward essential newborn care, in Guto Gida district, Oromia regional state, Ethiopia, September 2020.**

| Variables | Category | Frequency | Percentage |
|---|---|---|---|
| Have information on ENCP (n = 601) | Yes | 560 | 93.2 |
| | No | 41 | 6.8 |
| Knowledge on instruments used to cut the cord (n = 560) | Yes | 490 | 87.5 |
| | No | 70 | 12.5 |
| Knowledge of materials used to tie the cord (n = 560) | Yes | 514 | 85.5 |
| | No | 46 | 7.7 |
| Knowledge of substance applied to the cord (n = 560) | Yes | 401 | 66.7 |
| | No | 159 | 26.5 |
| Knowledge of bathing the newborn (n = 560) | Yes | 111 | 18.5 |
| | No | 449 | 74.7 |
| Knowledge of mothers on newborn breast feed (n = 560) | Yes | 111 | 18.5 |
| | No | 449 | 74.7 |
| Knowledge of mothers on newborn first feed (n = 560) | Yes | 499 | 83.0 |
| | No | 61 | 10.1 |
| Knowledge on ENCP (n = 560) | Good | 454 | 75.5 |
| | Poor | 106 | 17.6 |

danger signs out of the 6 common danger signs and were classified as having good knowledge of neonatal danger signs (Table 4).

## Components of essential newborn care practice

**Cord care practice.** The majority of respondents 443 (73.7%) cut their newborn cord with a new blade, 392 (65.2%) tied their cords with new thread, and 411 (68.4%) did nothing to their newborn cord (Table 5). Three hundred ninety-six (65.9%) of mothers had implement all essential cord care practice which were classified as good cord care practice, and 205 (34.1%) practiced poor cord care.

**Table 4. Knowledge status of home delivered women toward neonatal danger signs in Guto Gida district, Oromia regional state, Ethiopia, 2020.**

| Variables | category | Frequency | percentage |
|---|---|---|---|
| Awareness/information on neonatal danger signs (n = 601) | Yes | 477 | 78.4 |
| | No | 124 | 21.6 |
| Knowledge of mothers on poor sucking or not able feed breast (n = 477) | Yes | 371 | 61.7 |
| | No | 106 | 17.6 |
| Knowledge of mothers on Fast breathing (n = 477) | Yes | 283 | 47.1 |
| | No | 194 | 32.3 |
| Knowledge of mothers on hypothermia/babies feel cold (n = 477) | Yes | 83 | 13.8 |
| | No | 394 | 65.6 |
| Knowledge of mothers on febrile/fever (n = 477) | Yes | 346 | 57.6 |
| | No | 131 | 21.8 |
| Knowledge of mothers on drowsy or unconscious (n = 477) | Yes | 217 | 36.1 |
| | No | 260 | 43.3 |
| Knowledge of mothers on cord bleeding and infection (n = 477) | Yes | 134 | 22.3 |
| | No | 343 | 57.1 |
| Knowledge of women on neonatal danger signs (n = 477) | Good | 247 | 51.8 |
| | Poor | 230 | 48.2 |

**Table 5. Cord care practice among home delivered women, in Guto Gida district, Oromia regional state, Ethiopia, September 2020.**

| Variables | Category | Frequency | Percentage |
|---|---|---|---|
| Instruments used to cut the cord (n = 601) | New blade | 443 | 73.7 |
| | Old blade | 158 | 26.3 |
| Boiled old instruments used to cut the cord (n = 158) | Yes | 106 | 67.1 |
| | No | 52 | 32.9 |
| Materials used to tie the cord (n = 601) | New string/thread | 392 | 65.2 |
| | Old string/thread/Don't know/remember | 209 | 34.7 |
| Washing old materials used to tie the cord (n = 209) | Yes | 96 | 46.0 |
| | No | 113 | 5.0 |
| Applying substances to neonatal cord (n = 601) | Yes | 190 | 31.6 |
| | No | 411 | 68.4 |
| Substances those applied to neonate cord (n = 190) | Butter | 110 | 57.9 |
| | Vaseline/ointment/oil | 80 | 42.1 |

**Thermal care practice.** About 258 (42.9%) of women bathed their newborns after 24 hours of birth, and 325 (54.1%) of women dried and wrapped their newborns after delivery of the placenta (Table 6). Majority, 381(63.4%) of mothers had not implement all essential thermal care practice and classified as poor thermal care practice, and 220 (36.6%) practiced good thermal care.

**Breastfeeding practice.** With regard to breast-feeding practice, the majority 476 (79.2%) of participants provided first milk (colostrum) for their newborn, 395 (59.7%) of women initiated breastfeeding within 1 hour of birth, and 550 (91.5%) of women provided nothing for their newborn (Table 7). Majority, 355 (59.1%) of mothers had implement all essential breastfeeding practice, which was classified as good breastfeeding practice, and 246 (40.9%) practiced poor breastfeeding.

**Essential newborn care practice (ENCP).** The status of cord care practices, thermal care practices, and breastfeeding practices were assessed in this study, and the patterns of these three ENCP components were described by using percentages and frequencies. The prevalence of ENCP or women who practiced all three components of ENCP was 168 (28%) (23.9%-31.4%) (Fig 2).

## Predictors of essential newborn care practices

After successfully passing through the all-important statistical process, multivariable logistic regression analysis was conducted, and the results of the analysis was compared in terms of AORs, CIs, and p-value.

The study found that the wealth quintile was statistically significantly associated with essential newborn care practice. Mothers in the first wealth quintile were 36% less likely to practice good ENCP compared to mothers in the third wealth quintile (AOR [95% CI] = 0.64 [0.34–

**Table 6. Thermal care practice among home delivery women, in Guto Gida district, Oromia regional state, Ethiopia, September 2020, (n = 601).**

| Variables | Category | Frequency | Percentage |
|---|---|---|---|
| Bathing time of the newborn after delivery | within 1 hour | 59 | 9.8 |
| | Within 24 hours | 284 | 47.3 |
| | After 24 hours | 258 | 42.9 |
| Warping and drying of the newborn after delivery | Before delivery of placenta | 276 | 45.9 |
| | After delivery of placenta | 325 | 54.1 |

**Table 7. Breast feeding practice among home delivered women, in Guto Gida district, Oromia regional state, Ethiopia, September 2020, (n = 601).**

| Variables | Category | Frequency | Percentage |
|---|---|---|---|
| first breast milk (colostrum) was provided | Yes | 476 | 79.2 |
| | No | 125 | 20.8 |
| Breast feeding initiated within the first hour after delivery | Yes | 359 | 59.7 |
| | No | 242 | 40.3 |
| Provision of additional feed for the newborn | Yes | 51 | 8.5 |
| | No | 550 | 91.5 |

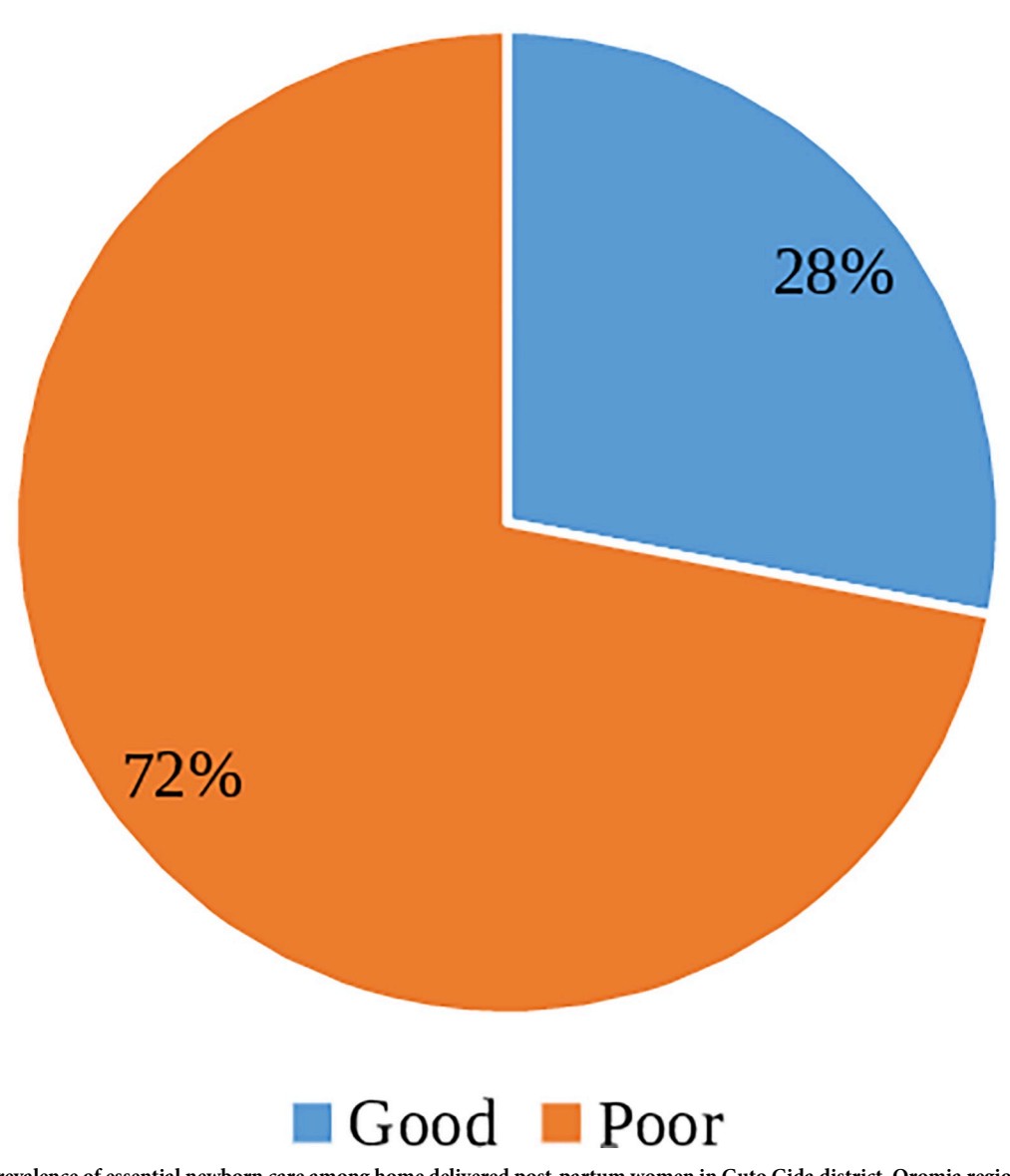

**Fig 2. Prevalence of essential newborn care among home delivered post-partum women in Guto Gida district, Oromia regional state, Ethiopia, May 2020.**

0.97]). Similarly, the number of live births (parity) was statistically significantly associated with ENCP. Women who had 1 alive birth were 49% less likely to practice good ENCP compared to women who had > = 5 live births (AOR [95% CI] = 0.51 [0.22–0.87]). The study also discovered that neonatal death was significantly associated with ENCP practice. Women who had lost their neonate prior to the study period were 89% less likely to practice good ENCP compared to their counterparts (AOR [95% CI] = 0.11 [0.05–0.22]).

In this study, women who had been visited and advised at their home by HEWs during pregnancy and PNC were 3.45 times more likely to practice good ENCP compared to those who did not receive ENCP advice during home visits (AOR [95% CI] = 3.45[1.56–7.26]). Similarly, women who received their ANC from skilled health professional during their current pregnancy were 1.79 times more likely to practice ENCP compared to their counterparts (AOR [95% CI] = 1.79 (1.21–3.36]). The study also discovered that women whose delivery was attended by HEWs were 3.29 times more likely to practice good ENCP than women who were attended by their relatives (AOR [95% CI] = 3.29 [2.13–5.94]) (Table 8).

## Discussion

It is interesting to note that this study was conducted in rural areas of Guto Gida district to assess the prevalence of ENCP and to identify determinant factors that hinder or promote ENCP among home-delivered mothers. The prevalence of ENCP was 168 (28%) (24.6%-31.8%) and the level of safe cord care, good breastfeeding, and good thermal care practices were (65.9%), (59.1%), and (36.6%), respectively. Our study findings also provided further evidence like: (42.9%) mothers bathed their newborn after 24 hours; (45.9%) mothers dried/wrapped the newborn immediately after birth; (59.7%) mothers initiated breast-feeding within one hour; and (79.2%) mothers provided colostrum for their newborn, which is lower than the WHO recommendation (100%).

In this study, 168 (28%) (24.6%-31.8%) of women demonstrated good ENCP, which is almost similar to previous studies conducted in Karnataka, India (28%) [19]. However, the current study finding is lower than the study conducted in rural area of South Ethiopia, Gedeo zone (24.1%) [7], study conducted in South Ethiopia Damot Pulasa (24%) [9], a study conducted in Chewaka Resettlement area, Ethiopia (5%) [11], a study conducted in Awabel district, East Gojam (23.1%) [20], and a study conducted in Ghana (2%) [5]. This difference may be due to multi-cultural variation among regions and the good status of mothers on pre-lacteal feeding, which has a rate of 91.5% in this study.

The prevalence of the current study is lower than previous studies conducted in Chencha District, South Ethiopia (38.4%) [4], Enderta Tigray, Ethiopia (40.7%) [8], Addis Ababa, Ethiopia (38.8%) [12], North West Ethiopia Mandura (41%) [21], and North Ethiopia Tigray (48.1%) [22]. This may be due to the low level of home visits by HEW (75% only visited), and the poor attitude of women on ENCP (approximately 66%) in the current study. The other possible reason is that, in the current study, all of the study participants were home-born without the help of skilled birth attendants, which contributed to the low prevalence of ENCP.

In this study, wealth quintile, number of live births (parity), neonatal death, home visit by HEW, ANC visit, and birth attendants were statistically significant predictors of ENCP.

In this study, women in the first wealth quintile were 36% less likely to practice ENCP as compared with women in the third wealth quintile. It was supported by the studies conducted in Chencha District, South Ethiopia, Eastern Uganda and rural areas of Nepal, which showed that women in the lower wealth quintiles were less likely to practice ENCP as compared to women in the highest wealth quintiles [4, 10, 23]. There are possible explanations for this finding. The first one is that women in the first wealth quintile were less likely to purchase a new

**Table 8. Multivariable logistic regression analysis results showing the predictors of ENCP among home delivery women, in Guto Gida district, Oromia regional state, Ethiopia, 2020.**

| Variables | Essential newborn care practice | | COR(95%CI) | AOR(95%CI) |
|---|---|---|---|---|
| | **Good** | **Poor** | | |
| **Age group** | | | | |
| 15–19 | 10(6.0%) | 73(16.9%) | 0.57(0.14–0.97) | 0.67(0.13–1.23) |
| 20–24 | 18(10.2%) | 76(17.5%) | 0.42(0.20–0.73) | 0.64(0.22–1.87) |
| 25–29 | 50(29.3%) | 90(20.8%) | 1.22(0.68–1.43) | 0.87(0.41–1.24) |
| 30–34 | 44(26.2%) | 97(22.4%) | 1.57(1.16–2.25) | 0.82(0.52–1.21) |
| 35–39 | 22(14.2%) | 76(17.6%) | 0.25(0.08–0.49) | 0.41(0.13–1.07) |
| 40–44 | 24(14.1%) | 21(4.8%) | 1.00 | 1.00 |
| **Educational status** | | | | |
| No formal education | 94(56.0%) | 314(72.5%) | 0.51(0.18–0.97) | 0.81(0.44–1.32) |
| Primary (1–8) | 60(35.7%) | 102(23.6%) | 0.71(0.37–0.92) | 0.74(0.25–1.46) |
| Secondary (9–12) | 14(8.3%) | 17(3.9%) | 1.00 | 1.00 |
| **wealth index in quintile** | | | | |
| 1st—quintile | 39(23.2%) | 148(34.2%) | 0.53(0.19–0.95) | 0.64(0.34–0.97) * |
| 2nd—quintile | 117(69.7%) | 237(54.7%) | 1.97(0.92–2.12) | 0.71(0.50–1.10) |
| 3rd—quintile | 12(7.1%) | 48(11.1%) | 1.00 | 1.00 |
| **Number of children (alive birth)** | | | | |
| One | 21(12.5%) | 131(30.2%) | 0.27(0.12–0.56) | 0.51(0.22–0.87) * |
| 2–4 | 127(75.6%) | 212(49.0%) | 2.69(0.92–4.81) | 1.43(0.78–2.05) |
| At > = 5 | 20(11.9%) | 90(20.8%) | 1.00 | 1.00 |
| **Visited and advised at home by HEWs** | | | | |
| Yes | 145(86.4%) | 308(71.1%) | 2.56(1.47–4.33) | 3.45 (1.56–7.26) * |
| No | 23(13.6%) | 125(28.9%) | 1.00 | 1.00 |
| **Attending ANC visit** | | | | |
| Yes | 155(92.3%) | 360(83.1%) | 2.42(1.23–5.01) | 1.79 (1.21–3.36) * |
| No | 13(7.7%) | 73(16.9%) | 1.00 | 1.00 |
| **Birth attendants** | | | | |
| HEWs | 78(46.4%) | 61(14.1%) | 4.50(1.09–7.29) | 3.29(2.13–5.94) * |
| TBAs | 73(43.4%) | 257(50.5%) | 1.92(0.84–4.42) | 2.20(0.73–4.42) |
| Mothers/sisters | 9(5.4%) | 79(18.1%) | 0.51(0.35–1.01) | 0.53(0.33–1.05) |
| Other relatives | 8(4.8%) | 36(8.3%) | 1.00 | 1.00 |
| **Neonatal status/condition** | | | | |
| Died | 4(2.4%) | 24(5.5%) | 0.42(0.10–0.67) | 0.11(0.05–0.22) * |
| Live | 164(97.6%) | 409(94.5%) | 1.00 | 1.00 |

COR = Crude odd ratio, AOR = Adjusted Odd Ratio

* = significant at p<0.05, 1 = Reference category, CI = Confidence Interval

blade to cut the cord, the new thread to tie the cord, and the clean clothes to warp the neonate. Another fact is that, when compared to women in the highest wealth quintile, women in the lowest wealth quintile were expected to have less access to information due to the unavailability of important items such as TVs, radios, and mobile phones in their homes.

In this study, women who had one live birth were 49% less likely to practice ENCP than women who had more than five live births. The reason for this association may be ongoing advice that has been given by health workers during repetitive pregnancy, delivery, and postnatal care. In contrast to this finding, the study conducted in Uganda found that multi-porous

mothers were 50% less likely to practice ENCP as compared to prime-porous mothers [23] This contradiction may be due to aged mothers' being more likely to prefer traditional practice than younger ones.

In this study, women who had home visits by HEWs and got advice on ENCP during pregnancy and PNC were 3.45 times more likely to practice ENCP as compared with women who had not been advised during home visits by HEWs. A consistent finding was documented by the study conducted in East Gojjam Awabel District, which found that women who had been advised on ENCP during their monthly visit were 4.8 times more likely to practice ENCP as compared with those women who were not advised during their monthly visit [20]. It was also supported by the studies conducted in four regions of Ethiopia, Addis Ababa, South Ethiopia (Damot Pulasa), and Nepal [6, 9, 12, 24]. The possible reason for this may be the ongoing progressive advice on ENCP during home visits by HEWs, which helps the mothers better understand and internalize the benefits of ENCP.

According to the study findings, women who had their ANC attended by a skilled health professional were 1.79 times more likely to practice ENCP than women who had not attended. This finding is supported by a study conducted in Chencha District of Southern Ethiopia [4], a study conducted in Enderta District, Tigray of Ethiopia [8], a study conducted in the rural districts of Gedeo zone of Southern Ethiopia [7], a study conducted in Mandura district of North West Ethiopia [21], a study conducted in Jimma [5], a study conducted in Eastern Uganda [23], a study conducted in Northern Ghana [17], a study conducted in Tanzania [25], and a study conducted in Sindhuli district of Nepal [10]. The possible reason may be the counseling that was conducted on the risk of poor ENCP and the benefit of good ENCP during ANC visit, which improved awareness of mothers on all activities of ENCP.

The current study found that women who had given birth by the help of HEWs were 3.29 times more likely to practice ENCP as compared with women who had given birth by the help of their relatives. This finding was supported by the studies conducted in Adis Abeba, Jimma, and Eastern Uganda [12, 23, 26]. This may be due to the fact that HEWs have more skill and experience on ENCP than other community members. However, this finding is inconsistent with a study done in Chencha District, South Ethiopia, [4]. The reason for this difference may be that some mothers reside in traditional mal-practices such as giving birth in the hands of more skilled birth attendants.

In this study, women who had lost their neonate before the study period were 89% less likely to practice ENCP as compared with women who had a live neonate. This finding confirms the fact that poor practicing of ENCP including the use of unclean cord cuts that may cause tetanus, the use of poor strength old cord ties that may cause cord bleeding and the application of substances to the cord that may cause infection can lead to neonatal death. The finding of the current study was contradicted by the studies conducted in Endenet Tigray Ethiopia, North West Ethiopia, and Northern Ghana [8, 17, 21]. The reason for this difference might be due to awareness creation made by health extension workers and other health care providers in case of the studies conducted in Endenet Tigray Ethiopia, North West Ethiopia, and Northern Ghana.

The present study used a cross-sectional study design; therefore, the temporal relationship between cause and effect was not addressed. The study used quantitative approach that makes unable to provide qualitative perspectives on essential newborn care practice. The other limitation is the use of only women's responses to conduct the study and the inability to consider bias that occurred as a result of differences between women's responses and actual observations. It study asked the women's past practice. As a result, recall bias may have affected the study.

Sample size was calculated for the prevalence of the outcome variable only, and not for factors associated with the outcome variable. This may lead to lack of adequate power and missing out on some other risk factors with small effect sizes.

## Conclusions and recommendations

The evidence from this study suggests that ENCP was demonstrated by less than one-third of women, safe cord care was demonstrated by two-thirds of women, good thermal care was demonstrated by about one-third of women, and good breastfeeding was demonstrated by two-thirds of women, which is lower than the WHO recommendation (100%). In this study, women in the first wealth quintile, women who had one live birth, and women who had lost their neonate before the study period were independent negative predictors for good ENC practice. Home visits by HEWs, ANC visits, and HEW attendance at birth were independent positive predictors of good ENC practice.

Healthcare planners should improve the current intervention plan on all components of ENCP and should give special attention to ANC service, home visits by HEW, women in lower wealth quintiles, and primi gravid (Para 1) women. Health office of the study district should organize an (IEC) program and enforce health care providers' recommendations in order to work efficiently. HWs should provide continued education and counseling services during ANC visits and delivery. HEWs should ensure they provide sufficient information on ENCP and neonatal danger signs during ANC visits, delivery, and home visits. They should give special attention to those women in the low-wealth quintile and primi gravid (Para one) during the ANC visit and home visit. Newborn care practice is a relatively crucial research topic and a government concern to achieve the sustainable development goal by 2030 and to reduce neonatal morbidity and mortality. Therefore, further research that uses advanced design and include qualitative part is needed.

## Supporting information

**S1 Questionnaire. English and Afan Oromo (Local language) versions questionnaire.**
(DOCX)

**S1 Data. SPSS data base.**
(SAV)

## Acknowledgments

The authors would like to thank Wollega University Institute of Health Sciences. Particular thanks also go to all supervisors, data collectors, and study participants for their cooperation and support during the study.

## Author Contributions

**Conceptualization:** Mulugeta Abebe, Melese Chego.

**Data curation:** Mulugeta Abebe, Melese Chego, Markos Desalegn.

**Formal analysis:** Mulugeta Abebe.

**Funding acquisition:** Mulugeta Abebe.

**Investigation:** Mulugeta Abebe, Gemechu Kejela, Markos Desalegn.

**Methodology:** Gemechu Kejela, Markos Desalegn.

**Project administration:** Markos Desalegn.

**Resources:** Melese Chego, Markos Desalegn.

**Software:** Melese Chego, Markos Desalegn.

**Supervision:** Mulugeta Abebe, Gemechu Kejela, Melese Chego, Markos Desalegn.

**Validation:** Gemechu Kejela, Melese Chego, Markos Desalegn.

**Writing – original draft:** Gemechu Kejela.

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
