## [Decision Letter · Decision Letter 0]

18 Aug 2022

PGPH-D-22-00752

Essential newborn care practices and associated factors among home delivered mothers in Guto Gida District, East Wollega Zone

Dear Dr. Kejela,

Thank you for submitting your manuscript to PLOS Global Public Health. After careful consideration, we feel that it has merit but does not fully meet PLOS Global Public Health’s publication criteria as it currently stands. Therefore, we invite you to submit a revised version of the manuscript that addresses the points raised during the review process.

Further, all reviewers and the editor feel that the formatting and language must improve to be able to communicate the message clearly in the manuscript. In the absence of such a revision, it may be difficult to evaluate the manuscript to the fullest. The authors are requested to correct the grammatical and typographical errors in the manuscript in the revised version. 

We look forward to receiving your revised manuscript.

Kind regards,

Ramachandran Thiruvengadam, M.D.,

Academic Editor

Journal Requirements:

1. Please upload all main figures as separate Figure files in .tif or .eps format only. For more information about how to convert and format your figure files please see our guidelines:

2. Please provide a/amend your detailed Financial Disclosure statement. This is published with the article. It must therefore be completed in full sentences and contain the exact wording you wish to be published.

3. Please provide separate figure files in .tif or .eps format.

4. Please amend your Data Availability Statement and indicate where the data may be found

Additional Editor Comments (if provided):

1. Abstract – In the following lines of results section, “Women who were advised on essential newborn care practice during a home visit by a health extension worker (3.45 p 0.01), women who attended antenatal care during their current pregnancy (1.79 p 0.01), and women who were attended at their birth by a health extension worker (3.29 p 0.01) were more likely to practice essential newborn care.”, the authors must provide the 95% CI for the association estimates and avoid p-values.

2. Methods section of the main manuscript – The authors write, “Women in the reproductive age group (15–49), who resided in the study area for at least six months, were reported to have delivered an alive baby within six months of postpartum and might have had an infant loss prior to the study period. mothers who were seriously ill, had a history of mental illness, and were unable to care for their infants, or care givers/guardians who cared for infants”. It seems that they are trying to describe the inclusion and exclusion criteria but these sentences aren’t conveying the information. Please rewrite.

3. Sample size calculation has been done to estimate the prevalence of the adoption of essential newborn care practices and not for the association analysis that they perform subsequently. This potentially could lead to lack of adequate power and missing out on the some other risk factors with small effect sizes. This should be mentioned in the limitation section.

4. The numbering of sections and subsections may be removed

5. The lists of dependent and independent variables may either be shifted to a table or written in to a narrative form, rather than bulleted lists.

6. Abbreviations need not be listed, better be avoided and if used, should be expanded at the first instance of usage.

7. In table-8, it is better to avoid p-values, if the author wants to present, then please give the actual p-vlaue instead of mentioning <0.01 etc.,

Reviewers' comments:

Reviewer's Responses to Questions

**Comments to the Author**

1. Does this manuscript meet PLOS Global Public Health’s publication criteria? Is the manuscript technically sound, and do the data support the conclusions? The manuscript must describe methodologically and ethically rigorous research with conclusions that are appropriately drawn based on the data presented.

Reviewer #1: Yes

Reviewer #2: Partly

Reviewer #3: Yes

2. Has the statistical analysis been performed appropriately and rigorously?

Reviewer #1: No

Reviewer #2: Yes

Reviewer #3: I don't know

3. Have the authors made all data underlying the findings in their manuscript fully available (please refer to the Data Availability Statement at the start of the manuscript PDF file)?

Reviewer #1: Yes

Reviewer #2: Yes

Reviewer #3: Yes

4. Is the manuscript presented in an intelligible fashion and written in standard English?

Reviewer #1: No

Reviewer #2: No

Reviewer #3: No

5. Review Comments to the Author

Reviewer #1: Abstract

• Methods

o No need to write about software details

o Focus on description of the protocol

• In the results section, expression values can be changed

o 95% CI instead of CI

o Write [ a vs.b ; p:0.01] format

• Conclusions

o the world health organization should be written as World Health Organization

o Recommendations can’t be in conclusion

o Only findings of the study need to be described

Introduction

• Looks a little lengthy

• Should focus on the basis for the study; shouldn’t comment about

• 2nd line: “the care they receive from delivery up to 28 days postpartum”

o Need to change it

o We are focusing on the neonatal care

o Authors can write it as till 28 days of life

o Postpartum period refers to maternal care

Methodology

• A little details about existing care available for the population can be mentioned

• Existing health care personnel manpower details can be mentioned

• WHO includes following in essential newborn care:

o Immediate care at birth (delayed cord clamping, thorough drying, assessment of breathing, skin-to-skin contact, early initiation of breastfeeding)

o Thermal care

o Resuscitation when needed

o Support for breast milk feeding

o Nurturing care

o Infection prevention

o Assessment of health problems

o Recognition and response to danger signs

o Timely and safe referral when needed

Authors need to mention the reason for choosing limited items

Results

• Expression of variable needs to be changed; should be according to the author’s guidelines

• Husband or spouse education details might help; Also highest educational status in the family will also help in understanding the mother’s family status. Also we can utilize their knowledge to look the ENC practices; If author’s have the details, they can mention

•

Discussion

• Authors should try to describe and compare the results with other studies..No description of other study details and values. Also authors should tell how the present study fares against them in terms of strengths and limitations

• Limitations

o The design of the study has inherent limitations; Authors need to mention it

o Recall bias also need to be mentioned

Typo errors and grammar needs to be corrected; 3 level heading policy doesn't require authors to have the numbering; It can be removed.

Authors can write a footnote below each table: need to expand abbreviations used in the table;

Reviewer #2: Introduction:

The introduction explains the importance of Essential newborn care. The justification for the study is written in a lengthy way which could be shortened.

Methods and Materials:

The primary and secondary objectives of the study were not explicitly mentioned.

Population:

Inclusion and exclusion criteria were not mentioned clearly.

Sampling methods and study design were explained in detail.

Operational definitions:

Dependent variables:

Three variables and their scoring methods could be explained in simple words or table for easy clarity.

Poor ENCP was also assigned a score of 1. (? Typographical error)

Independent variables:

There is a mention about 12 independent variables for ENCP. Those could be numbered and clearly mentioned.

The scale used to measure wealth index to be mentioned

Results:

The description of data mentioned in tables could be avoided.

Only 560 of the total 601 study participants were assessed for knowledge on ENCP. The reasons were not mentioned.

Similarly, only 477 were evaluated for knowledge on neonatal danger signs.

All the independent variables used and the results of the analysis for each variable could be mentioned.

Though used as independent variables, knowledge and attitude on ENCP was not used for analysis.

The data collected for reasons for home delivery were not used in analysis. The importance of those data for the study objectives has to be mentioned.

Discussion:

The paragraph mentioning the association of neonatal loss with lack of ENCP is difficult to understand. Reframing sentences for clear understanding is suggested.

Tables:

Table 2: The frequency of subjects advised on handwashing, drying baby, breastfeeding and those who were not advised do not add to the total of 601 subjects.

Tables 3 & 4: The number of respondents for each category is varying and not all study subjects data were not studied.

Reviewer #3: This study is the valid and valuable confirmation of the state of essential newborn care practices in rural Ethiopia. Authors have used rigorous research tools to test the research question. However authors need to pay attention to the English as some of the sentences need grammatical correction so that they can be meaningful. For example the section on population in section 2.2 under methods- does not convey any meaning.

6. PLOS authors have the option to publish the peer review history of their article (what does this mean?). If published, this will include your full peer review and any attached files.

**Do you want your identity to be public for this peer review?** For information about this choice, including consent withdrawal, please see our Privacy Policy.

Reviewer #1: No

Reviewer #2: No

Reviewer #3: **Yes: **Chandra Kumar Natarajan

---

## [Decision Letter · Decision Letter 1]

25 Oct 2022

PGPH-D-22-00752R1

Essential newborn care practices and associated factors among home delivered mothers in Guto Gida District, East Wollega Zone

Dear Dr. Kejela,

Thank you for submitting your manuscript to PLOS Global Public Health. After careful consideration, we feel that it has merit but does not fully meet PLOS Global Public Health’s publication criteria as it currently stands. Therefore, we invite you to submit a revised version of the manuscript that addresses the points raised during the review process.

We look forward to receiving your revised manuscript.

Kind regards,

Ramachandran Thiruvengadam, M.D.,

Academic Editor

Journal Requirements:

Additional Editor Comments (if provided):

Reviewers' comments:

Reviewer's Responses to Questions

**Comments to the Author**

1. If the authors have adequately addressed your comments raised in a previous round of review and you feel that this manuscript is now acceptable for publication, you may indicate that here to bypass the “Comments to the Author” section, enter your conflict of interest statement in the “Confidential to Editor” section, and submit your "Accept" recommendation.

Reviewer #1: All comments have been addressed

Reviewer #2: All comments have been addressed

2. Does this manuscript meet PLOS Global Public Health’s publication criteria? Is the manuscript technically sound, and do the data support the conclusions? The manuscript must describe methodologically and ethically rigorous research with conclusions that are appropriately drawn based on the data presented.

Reviewer #1: No

Reviewer #2: Yes

3. Has the statistical analysis been performed appropriately and rigorously?

Reviewer #1: Yes

Reviewer #2: Yes

4. Have the authors made all data underlying the findings in their manuscript fully available (please refer to the Data Availability Statement at the start of the manuscript PDF file)?

Reviewer #1: Yes

Reviewer #2: Yes

5. Is the manuscript presented in an intelligible fashion and written in standard English?

Reviewer #1: Yes

Reviewer #2: Yes

6. Review Comments to the Author

Reviewer #1: Spellings of sections headers like “conclusion”, “Introduction”, “Key word” and “study” needs correction. Many needs correction in text, tables and figures

Methodology:

The detailing of “Yes/No and 1/0” can be removed as you have mentioned definitions. Keep categorizations only when authors have more than >2 categories exist.

Sample size formula can be removed

Reviewer #2: All comments have been addressed. The language of the manuscript is now easy to understand.

7. PLOS authors have the option to publish the peer review history of their article (what does this mean?). If published, this will include your full peer review and any attached files.

**Do you want your identity to be public for this peer review?** For information about this choice, including consent withdrawal, please see our Privacy Policy.

Reviewer #1: No

Reviewer #2: No

---

## [Decision Letter · Decision Letter 2]

14 Dec 2022

Essential newborn care practices and associated factors among home delivered mothers in Guto Gida District, East Wollega Zone

PGPH-D-22-00752R2

Dear Dr Kejela,

We are pleased to inform you that your manuscript 'Essential newborn care practices and associated factors among home delivered mothers in Guto Gida District, East Wollega Zone' has been provisionally accepted for publication in PLOS Global Public Health.

Best regards,

Ramachandran Thiruvengadam, M.D.,

Academic Editor

Reviewer Comments (if any, and for reference):

Reviewer's Responses to Questions

**Comments to the Author**

1. If the authors have adequately addressed your comments raised in a previous round of review and you feel that this manuscript is now acceptable for publication, you may indicate that here to bypass the “Comments to the Author” section, enter your conflict of interest statement in the “Confidential to Editor” section, and submit your "Accept" recommendation.

Reviewer #1: All comments have been addressed

2. Does this manuscript meet PLOS Global Public Health’s publication criteria? Is the manuscript technically sound, and do the data support the conclusions? The manuscript must describe methodologically and ethically rigorous research with conclusions that are appropriately drawn based on the data presented.

Reviewer #1: Yes

3. Has the statistical analysis been performed appropriately and rigorously?

Reviewer #1: Yes

4. Have the authors made all data underlying the findings in their manuscript fully available (please refer to the Data Availability Statement at the start of the manuscript PDF file)?

Reviewer #1: Yes

5. Is the manuscript presented in an intelligible fashion and written in standard English?

Reviewer #1: Yes

6. Review Comments to the Author

Reviewer #1: Authors have done the necessary corrections

7. PLOS authors have the option to publish the peer review history of their article (what does this mean?). If published, this will include your full peer review and any attached files.

**Do you want your identity to be public for this peer review?** For information about this choice, including consent withdrawal, please see our Privacy Policy.

Reviewer #1: No
